# Laboratory Calibration of D-dot Sensor Based on System Identification Method

**DOI:** 10.3390/s19153255

**Published:** 2019-07-24

**Authors:** Ke Wang, Yantao Duan, Lihua Shi, Shi Qiu

**Affiliations:** National Key Laboratory on Electromagnetic Environment Effects and Electro-Optical Engineering, Army Engineering University of PLA, Nanjing 210007, China

**Keywords:** electromagnetic pulse measurements, D-dot sensor, system identification, frequency–domain calibration

## Abstract

D-dot sensors can realize the non-contact measurement of transient electric fields, which is widely applied to electromagnetic pulse (EMP) measurements with characteristics of the wide frequency band, high linearity, and good stability. In order to achieve accurate calibration of D-dot sensors in the laboratory environment, this paper proposed a new calibration method based on system identification. Firstly, the D-dot sensor can be considered as a linear time-invariant (LTI) system under corner frequency, thus its frequency response can be characterized by the transfer function of a discrete output error (OE) model. Secondly, based on the partial linear regression of the transfer function curve, the sensitivity coefficient of the D-dot sensor is obtained. By increasing the influence weight of low-frequency components, this proposed method has better calibration performance when the waveform is distorted in the time domain, and can artificially adapt to the operating frequency range of the sensor at the same time.

## 1. Introduction

The measurement of the transient electric fields is an important part of electromagnetic pulse (EMP) protection research. The purpose of the measurement is to determine the EMP environment, the response to EMP and the protective effect of the equipment and system under test [1]. The D-dot sensor is widely applied to EMP measurements for its wide frequency band, high linearity, and good stability [2,3,4,5]. This passive non-contact measurement method is simple to implement and easy to arrange with a small distortion to the measured field. Moreover, the sensitivity coefficient can be calculated theoretically and only related to the structure of the probe, so there is no need to repeat the calibration periodically, which greatly facilitates the measurement [6].

C. E. Baum summed up the work of his predecessors, revealed the working principle of D-dot sensors and gave the theoretical calculation formula of the equivalent capacitance in. However, due to the introduction of stray capacitance, which reduces the sensitivity of D-dot sensor, accurate calculation of the sensitivity coefficient is difficult to implement. In addition, the theoretical calculation is also limited by the shape of probe [7,8,9].

There is a difference between the theoretical model and the actual situation, so any measurement system must be experimentally calibrated. Through calibration, we can obtain the pulse response index, sensitivity, response bandwidth, dynamic range and nonlinear distortion of the D-dot sensor. Therefore, the D-dot sensor should be calibrated in a laboratory environment. Because of the complex electromagnetic environment of the laboratory, the calibration waveforms are often disturbed by noise resulting in errors of calibration coefficient, especially the overshoot or ringing occurred at the peak. This paper proposed a frequency–domain calibration method based on system identification, which has better calibration performance when dealing with time domain waveform distortion, and can adapted to the signal bandwidth at the same time.

## 2. Theoretical Basis

### 2.1. D-dot Sensor

Considering the Norton equivalent circuit shown in Figure 1, the antenna of D-dot sensor can be regarded as a current source, and current in the time domain can be written:(1)$$i_{s}=C\frac{du_{0}(t)}{dt}+\frac{u_{0}}{R}$$

Then output voltage *u*_0_ can be represented in the frequency domain as:(2)$$U_{0}(j\omega )=\frac{j\omega DA_{e}R}{j\omega RC+1}$$

For a particular D-dot sensor, characteristic impedance *R* and equivalent capacitance *C* can be regarded as constants; then the amplitude–frequency response can be expressed as a curve that varies with frequency, as shown in Figure 2. According to the different frequency response characteristics, D-dot sensors can be divided into three working modes: the first-order differential mode (*f* < *f*_3d_), the transition mode (*f*_3d_ < *f* < *f*_3i_), and the self-integration mode (*f* > *f*_3i_).

The first-order differential mode, also known as D-dot mode, is the main working mode in which the amplitude linearity attenuation does not exceed 0.3 dB. Therefore, the output voltage *u*_0_ of D-dot sensor can be considered to be proportional to the incident electric field rate of change in D-dot mode. Usually, the corner frequency *f*_0_ of the D-dot sensor is designed to be several hundred megahertz, so that the main spectral components of the EMP to be measured are below this frequency [8].

### 2.2. System Identification Calibration Method

Since the D-dot probe satisfies the homogeneity and superposition criteria in D-dot mode, and the output-to-input relationship has a shift-invariant characteristic, it can be regarded as a linear time-invariant (LTI) system in the D-dot mode; then a digital filter in series with the input signal can be designed to forecast the output response of the system, namely the system transfer function of D-dot sensor.

Consider the discrete output error (OE) model [10], given as follows:(3)$$y(n)=H(z^{-1})x(n)+v(n)$$
where *x*(*n*) is the system input sequence, *y*(*n*) is the system output sequence, *v*(*n*) is stochastic white noise sequence with zero mean and variance *σ*^2^ = 0, and the system transfer function is defined as:(4)$$H(z^{-1})=\frac{B(z^{-1})}{F(z^{-1})}=\frac{b_{0}+b_{1}z^{-1}+\cdots +b_{n_{b}}z^{-n_{b}}}{1+f_{1}z^{-1}+\cdots +f_{n_{f}}z^{-n_{f}}}$$
where *B*(*z*^−1^) and *F*(*z*^−1^) are polynomials in the unit backward shift operator *z*^−1^, *z*^−*i*^*x*(*n*Δ*t*) = *x*[(*n* − *i*)Δ*t*], Δ*t* is the sampling interval.

Assuming that the slope of the amplitude–frequency response curve in D-dot mode is *k*, the transfer function can be written as:(5)$$\left|H(j\omega )\right|=\frac{k}{2\pi }\omega$$

After eliminating stochastic white noise, the relation of the system can be approximately expressed as:(6)$$y(t)=\frac{k}{2\pi }\cdot \frac{dx(t)}{dt}$$

Since the vertical electric field of TEM (transverse electromagnetic) cell is uniformly distributed, the actual incident electric field rate of change *E’* can be obtained by:(7)$$E^{\prime }(t)=\frac{dx(t)}{hdt}$$
where *h* refers to the height difference between the core plate and the D-dot sensor. Then the sensitivity coefficient *k*_s_ of D-dot sensor can be represented as:(8)$$k_{S}=\frac{E^{\prime }(t)}{y(t)}=\frac{2\pi }{h\cdot k}$$

## 3. Calibration Procedures

The system identification calibration method can be divided into four steps: experimental design, pretreatment, model parameter identification, and partial linear regression.

### 3.1. Experimental Design

As illustrated in Figure 3, an exclusive calibration platform for D-dot sensor was built in the laboratory [11], which mainly composed of an excitation source, a TEM cell, and a shielding room.

The excitation source was selected the PRIMA EFT61004B pulse group generator, which can output a 5/50 ns double-exponential impulse voltage. At the same time, it could achieve a change of amplitude of output voltage from 0.2 kV to 2 kV. From the perspective of spectral analysis, the bandwidth of the excitation source should be sufficient to cover the recognized system. For calibration, the ideal signal source should be as narrow as possible in the time domain, such as unit-impulse, but for system identification, it means the sampling rate has to increase dramatically. According to [12], if the dominant time constant of the system is *T*_m_, when the sampling time *T*_0_ = 10 *T*_m_, the identification accuracy decreases by 10^5^ times. Therefore, a double exponential narrow pulse is the appropriate selection of different demands for the source signal when calibrating and identifying.

The TEM cell was used to generate transverse electromagnetic waves, which has an input measurement port and an output measurement port both matched the coaxial connectors of 50 ohms. The vertical electric fields in the TEM cell can be considered to be evenly distributed, so the value of the electric field is approximately equal to the voltage between the core and bottom plate divided by the height *h*, *h* = 0.5 m. Therefore, the EMP waveform in the vicinity of the D-dot sensor, placed at the center of the bottom plate, can be calculated from the output waveform of the TEM cell.

The oscilloscope, Tektronix MDO3024, is placed at the shielding room to minimize the interference of environmental noise, resulting in improved signal-to-noise ratio (SNR) of the acquired signal. In addition, the laboratory independently designed a D-dot sensor with a cone probe to realize the measurement of the fast-front EMP, and its physical diagram is shown in Figure 4 [13].

### 3.2. Pretreatment

Since the direct application of observed data to the identification will greatly reduce the identification accuracy and the estimation ability of the model order, in the process of signal pretreatment, resampling, detrending, denoising and time delay elimination are processed in turn.

The choice of sampling time will directly affect the accuracy of the identification model. Too long will lead to a lack of information, too short will reduce the estimation of static gain [10]. When the oscilloscope initially collects data, oversampling is usually used to ensure the integrity of the waveform information, so resampling processing is needed to reduce the quantization noise, so it was reset at 1GHz in this paper.

Assuming that the observed input data *x**(*n*) and output data *y**(*n*) contains a DC component, and the detrended data is recorded as:(9)$$\{\begin{array}{l} x(n)=x^{*}(n)-x_{0}\\ y(n)=y^{*}(n)-y_{0} \end{array}$$ where *x*_0_ and *y*_0_ are the DC components of the input and output, respectively, and the following recursive estimation method can be used [14]:(10)$$\{\begin{array}{l} {\hat{x}}_{0}(n)={\hat{x}}_{0}(n-1)+\frac{1}{n}[x^{*}(n)-{\hat{x}}_{0}(n-1)]\\ {\hat{y}}_{0}(n)={\hat{x}}_{0}(n-1)+\frac{1}{n}[y^{*}(n)-{\hat{x}}_{0}(n-1)] \end{array}$$
where *x*(*n* − 1) and *y*(*n* − 1) represent the estimated value of the DC component at the previous moment.

The Savitzky–Golay filter (usually referred to as S–G filter) was originally proposed by Savitzky and Golay in 1964. It is a filtering method based on local polynomial least squares fitting in the time domain [15]. The most important feature of this filter is that it can keep the shape and width of the signal unchanged while filtering out the noise. Suppose that the width of the filter window is 2*m* + 1, *x*(*i*) is a set of data in *x*(*n*), *i*= −*m*, ..., 0, ..., *m*, and after smoothing the *x*(*i*) through S–G filter, *X*(*i*) is obtained:(11)$$X(i)=\sum_{k=-m}^{m}h(k)x(i-k)$$ where *h*(*k*) is the sampling response of S–G filter. Similarly, the output data *y*(*n*) can be smoothed.

Due to the difference in transmission distance, there is a time delay between the output and the input signal. For the sake of reducing the identification variables, the cross-correlation method is used to remove the relative time delay. From Figure 5, the delay step corresponding to the peak value is the time delay estimation value, here are 15 sample points, that is, the output signal needs to be shifted forward by 15 time-steps to remove this phenomenon.

### 3.3. Model Parameter Identification

For the OE model established in Equation (4), the parameters to be estimated can be written as:(12)$$\theta ={\left[b_{0}\ldots b_{n_{b}}f_{1}\ldots f_{n_{f}}\right]}^{T}$$ where the order *n*_b_ and *n*_f_ were determined by the Akaike information criterion (AIC), which is a standard to measure the goodness of fit of the statistical model [16]. It can be determined by:(13)$$\mathrm{AIC}(N_{\theta })=-2\ln L({\hat{\theta }}_{\mathrm{ML}})+2N_{\theta }$$ where ${\hat{\theta }}_{\mathrm{ML}}$ refers to the maximum likelihood estimate of model parameters, $L({\hat{\theta }}_{\mathrm{ML}})$ refers to maximum likelihood function under parameter estimation condition.

Assuming that the maximum order of *n*_b_ and *n*_f_ is 5, and omitting input delay, *n*_k_ = 0. The calculation results of Akaike information criterion (AIC) under different order are illustrated in Table 1.

AIC has a minimum if and only if *n*_b_ = 3 and *n*_f_ = 3, the system transfer function can be determined, such as:(14)$$H(z^{-1})=\frac{0.00651791-0.00126895z^{-1}-0.00523787z^{-2}}{1+1.06972z^{-1}+0.466039z^{-2}-0.108918z^{-3}}$$

Model output was obtained by the measured input which was used as the model input, and the comparison of model output and measured output is shown in Figure 6. It can be concluded that the model output is basically consistent with the measured output, reflecting the response and distortion of the system under the excitation source, and verifying the validity of the model. In Figure 6, the fitting degree of two curves is approximately 87.77%.

In addition, 30 sets of calibration data with input pulse voltage amplitude ranging from 200 V to 2 kV are randomly selected to verify the universality of the identification model. Finally, the average fitting degree is more than 85% by statistics, which shows good generality.

### 3.4. Partial Linear Regression

The OE model can provide a parametric description of the transfer function, it is much clearer than the conventional frequency domain curves. Firstly, according to Figure 2, the cut-off frequency of D-dot mode *f*_3d_ can be obtained by linearity criterion—the attenuation does not exceed 0.3 dB. Secondly, as illustrated in Figure 7, the slope of the amplitude–frequency response curve in D-dot mode is represented by the partial linear regression of the transfer function. Thirdly, the sensitivity coefficient can be solved by Equation (9).

At the same time, the frequency domain calibration method can flexibly adapt operating frequency range of the D-dot sensor, and only needs to adjust the frequency range of the linear regression. Thus, the system identification method can increase the influence weight of low-frequency components when the time–domain signal suffers a waveform distortion.

## 4. Results and Discussion

According to Equation (2), the sensitivity coefficient *k*_S_ also can be theoretically determined by:(15)$$k_{s}=\frac{1}{\varepsilon_{0}A_{e}R}$$

Then the equivalent area of cone probe antenna can be determined by:(16)$$A_{e}=\frac{Ch_{e}}{\varepsilon_{0}}\sqrt{1-\tan(\theta /2)}$$
where *C* refers to the equivalent capacitance of the probe antenna, *C* = 3.6 pF; *h*_e_ refers to the height of the cone probe, *h*_e_ = 2.7 cm. The characteristic impedance of an infinitely long ideal conductor cone probe *Z*_c_ is determined only by its half cone angle *θ*. In order to match the impedance of a 50 ohms transmission cable, *θ* is taken as 47° [13].
(17)$$Z_{c}=60\ln(\cot\frac{\theta }{2})$$

The simultaneous Equations (16) and (17) are available: the theoretical calculation result of sensitivity coefficient approximately is 2.74 × 10^11^ (V/m/s)/V, which can be used as a reference for laboratory calibration.

### 4.1. Laboratory Method Comparison Results

As shown in Figure 8, there is a typical waveform distortion in the calibration data obtained on the platform shown in Figure 3. There is a second peak appeared after the peak, which means that the calibration data may have undergone time domain distortions. At this point, the result of the peak value calibration method will not be trusted and need to verify. Using the data shown in Figure 8, the calibration results are eventually illustrated in Table 2. What’s worth mentioning, each result is an average of three calibrations in the table, and the time–domain calibration method and frequency–domain calibration method are used respectively.

The peak value calibration method is a common time–domain calibration method, which solves the sensitivity coefficient by the ratio of the peak value between the input and output. The frequency–domain calibration method can be divided into the FFT method and system identification method depending on how to obtain the transfer function.

From Table 2, the results of the peak value calibration method and the system identification method are 2.61 × 10^11^ and 2.70 × 10^11^ (V/m/s)/V, respectively, and the difference between them is within 5%. When a faster signal is driven along a long transmission line and there is no effective matching at the same time, it often occurs that the electric field waveform distorted by a reflected wave, which eventually effects results by time–domain calibration method.

Comparing Figure 7 with Figure 9, the transfer function solved by FFT method and the transfer function solved by system identification method has very high coincidence below 50 MHz. However, as the frequency increases, the transfer function curve of the FFT method oscillates resulting in a sharp drop in calibration accuracy. Due to the oscillation of the transfer function curve of the FFT solution, the corner frequency is more difficult to determine and the calibration accuracy is poorer. Therefore, the system identification method can be considered as an improvement of the FFT method, which can give a smooth curve of the transfer function.

### 4.2. Calibration Results Verification

In order to verify the sensitivity coefficient calibrated by the system identification method, the experimental verification was carried out in the laboratory.

The setup of the verification experiment was based on the calibration platform shown in Figure 3. However, there are several differences in the experimental setup that needs to be pointed out. First, the SHANGHAI SANKI ENS-24XA high-frequency noise generator was used as an excitation source to generate the square wave. Second, due to the bandwidth limitation of the TEM cell, the square wave verification experiment carried out on the basis of a GTEM (giga hertz transverse electromagnetic) cell. Third, due to the different structure of the GTEM cell, the incident electric field was measured by a high voltage probe, which is placed at the input port of the GTEM cell. Four, the relative delay of the input and output waveforms is removed to facilitate comparison.

The verification results are shown in Figure 10, where the blue line refers to the incident electric field measured by the high voltage probe; the red represents to the waveform of numerical integration multiplied by the sensitivity coefficient calibrated by system identification method, 2.70 × 10^11^ (V/m/s)/V, which is measured by the D-dot sensor. Comparing the coefficients of time–domain calibration and frequency–domain calibration, it is obvious that the latter is closer to the incident electric field.

## 5. Conclusions

This paper proposed a frequency–domain calibration method of D-dot sensors on the basis of system identification. The system transfer function was utilized to approximate the frequency response characteristics of the D-dot sensor, then the sensitivity coefficient adapted the operating frequency range can be obtained by partial linear regression. This method can adjust the weight of influence of low-frequency components, which are more stable. In comparison, the system identification method has higher accuracy on calibration than traditional time–domain method when the signal waveform suffers a distortion, such as overshoot and ringing, and it can provide a parametric description of the transfer function and much clearer frequency domain curves than FFT method. In short, there are more application scenarios than the two traditional methods.

In future research, the proposed method has certain application prospects in the calibration of linear systems. For example, when the waveform distortion of nuclear electromagnetic pulse sensor is serious and the time domain calibration method is no longer applicable, the system identification method can be a better solution.

## Figures and Tables

**Figure 1 sensors-19-03255-f001:**
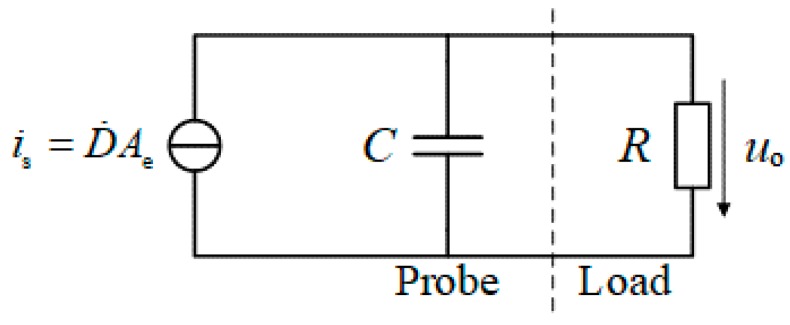
Norton Equivalent circuit of D-dot sensor. Where *D* refers to the electric flux density of the measured electric field; *C* refers to the equivalent capacitance of the D-dot sensor; *R* refers to the characteristic impedance of the transmission line; *u*_0_ refers to the output voltage of the D-dot sensor; *A*_e_ refers to the equivalent area of probe antenna.

**Figure 2 sensors-19-03255-f002:**
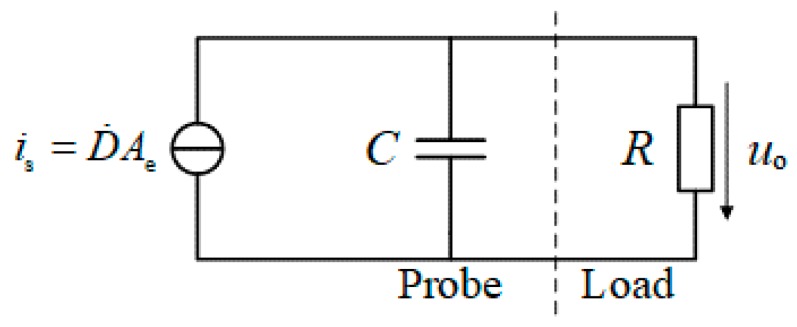
Amplitude–frequency response curve of D-dot sensor.

**Figure 3 sensors-19-03255-f003:**
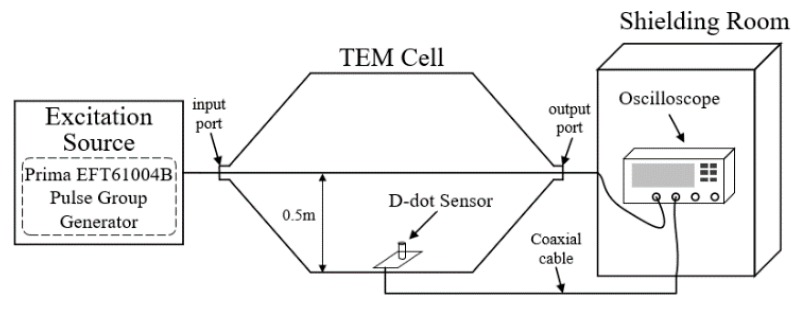
Schematic of calibration platform for D-dot sensors.

**Figure 4 sensors-19-03255-f004:**
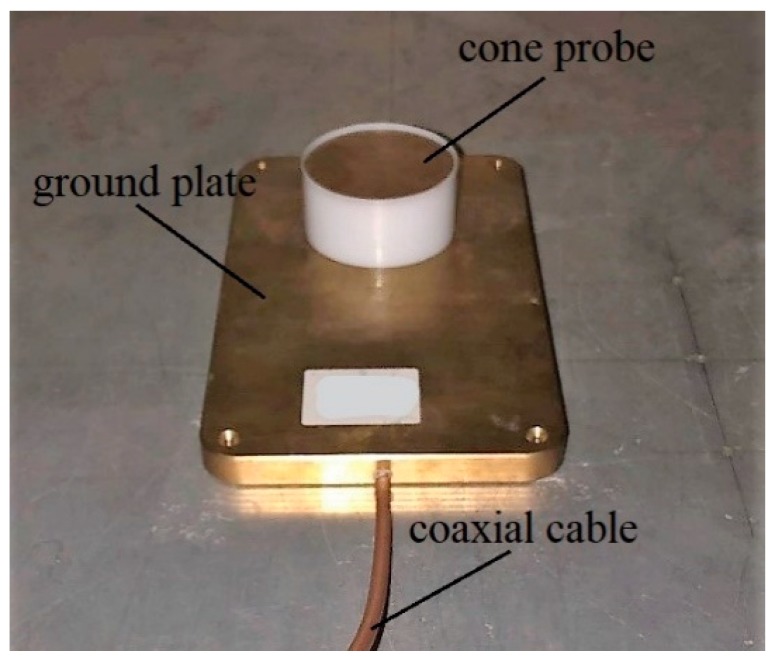
Physical diagram of D-dot sensor. Where the half cone angle of the probe is 47°.

**Figure 5 sensors-19-03255-f005:**
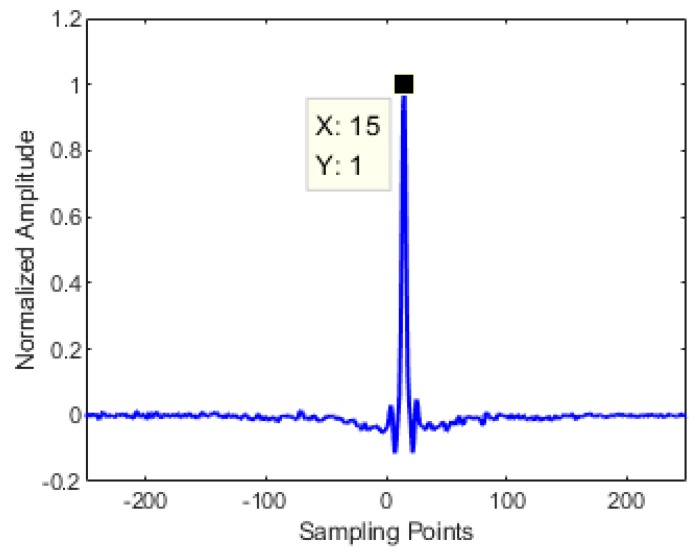
Time delay estimation by cross-correlation method.

**Figure 6 sensors-19-03255-f006:**
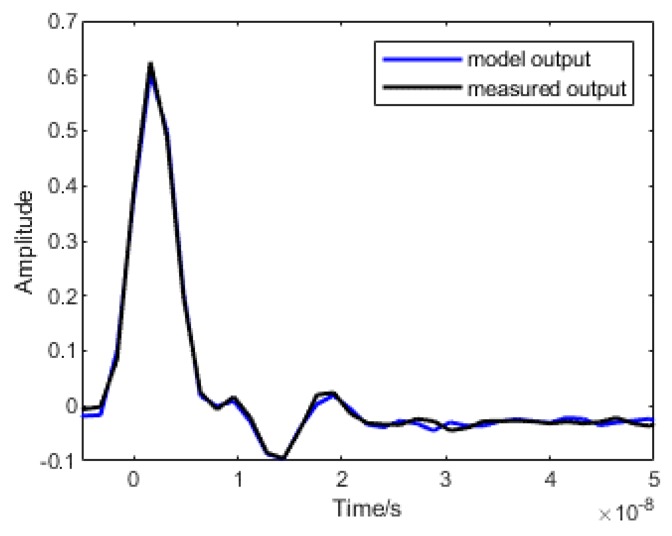
The verification of the output error model. Where the blue line refers to the model output, the black line refers to measurement output.

**Figure 7 sensors-19-03255-f007:**
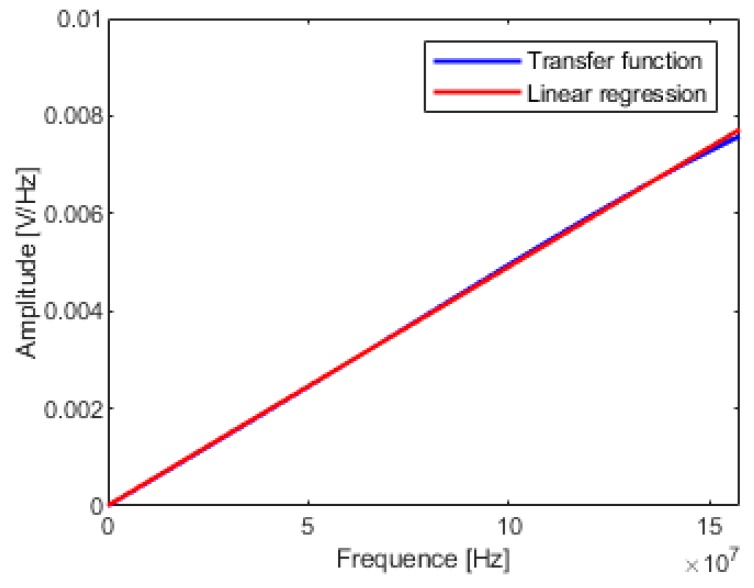
Partial linear regression of transfer function.

**Figure 8 sensors-19-03255-f008:**
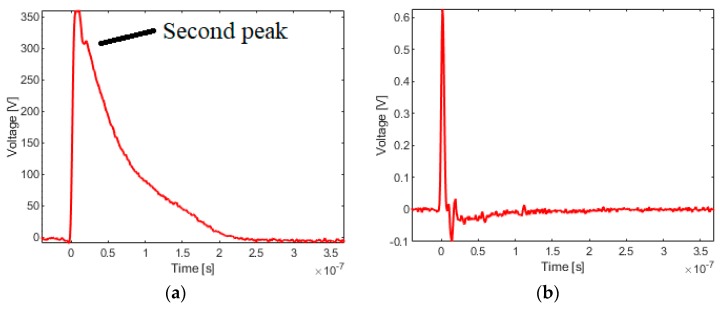
A set of calibration data (**a**) Input signal. (**b**) Output signal.

**Figure 9 sensors-19-03255-f009:**
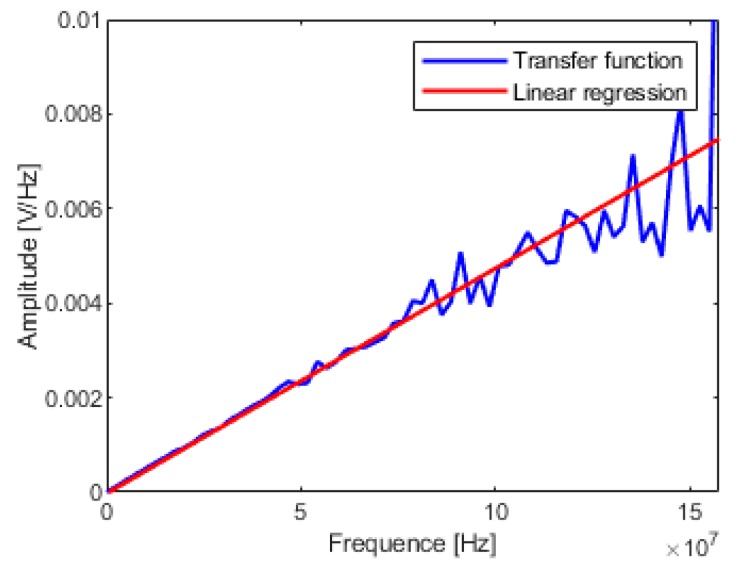
Partial linear regression of the FFT method.

**Figure 10 sensors-19-03255-f010:**
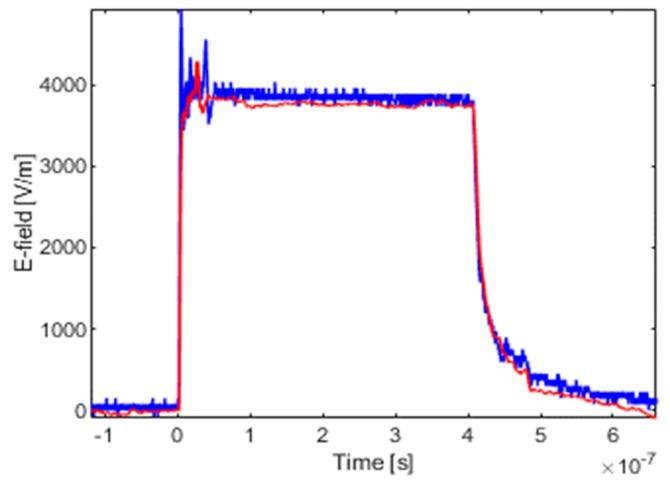
Experimental verification of square wave.

**Table 1 sensors-19-03255-t001:** Akaike information criterion under a different model order.

AIC (×10^3^)		*n* _f_				
		1	2	3	4	5
***n*_b_**	**1**	−1.5384	−	−	−	−
	**2**	−2.2234	−2.2291	−	−	−
	**3**	−2.2275	−2.2285	−2.2294	−	−
	**4**	−2.2280	−2.2281	−2.2267	−2.2248	−
	**5**	−2.2292	−2.2267	−2.2247	−2.2260	−2.2291

**Table 2 sensors-19-03255-t002:** Comparison of sensitivity coefficient calibration methods.

U ^1^ [kV]	*k*_S_ [(V/m/s)/V]		
	Peak Value	FFT	System Identification
0.4	2.64 × 10^11^	2.55 × 10^11^	2.67 × 10^11^
0.6	2.57 × 10^11^	2.72 × 10^11^	2.78 × 10^11^
0.8	2.61 × 10^11^	2.38 × 10^11^	2.72 × 10^11^
1	2.57 × 10^11^	2.89 × 10^11^	2.69 × 10^11^
1.2	2.61 × 10^11^	2.37 × 10^11^	2.72 × 10^11^
1.4	2.64 × 10^11^	2.71 × 10^11^	2.71 × 10^11^
1.6	2.61 × 10^11^	2.89 × 10^11^	2.71 × 10^11^
1.8	2.65 × 10^11^	2.76 × 10^11^	2.67 × 10^11^
2	2.56 × 10^11^	2.94 × 10^11^	2.64 × 10^11^
Average	2.61 × 10^11^	2.69 × 10^11^	2.70 × 10^11^
Stdev ^2^	3.35 × 10^9^	2.14 × 10^11^	4.01 × 10^9^

^1^ U refers to the charge voltage of the pulse group generator. ^2^ Stdev refers to the standard deviation of sensitivity coefficients.

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
