# Peer review of "Laboratory Calibration of D-dot Sensor Based on System Identification Method"

_sensors, 2019, doi:10.3390/s19153255_

Reviewer 1 Report

see comments on pdf file

Author Response

Response to Reviewer 1 Comments

Thank you very much for your extensive work on reviewing our manuscript. We have revised the paper according to your valuable comments. The revised parts of the new manuscript are marked and the problems are addressed below sequentially.

Point 1: Some syntax and formatting errors pointed out.

Response 1: We have revised it in the manuscript.

Point 2: the results supported is not enough to support the conclusions

Response 2: The manuscript has been improved.

Reviewer 2 Report

The paper is in principle interesting and probably worth publishing.

However, there are a number of things that needs to be clarified.

What will be the final use of the D-dot probe? Will it be used as an open field sensor?

The variation of the D-dot sensor with frequency (see Fig. 1 and Eq. 2) is only valid when the sensor is mounted inside a metallic structure, in respect to which there is a capacitance of the dome. If the sensor is used as an open field sensor this is no more the case.

When the calibration experiments are performed to obtain the calibration factor, there are a large number of parameters that have manufacturing or/and measurement errors that influence the results. For example h in Eq.7, he, C and theta in eq 16 are all are known with a certain error. When ks is calculated, these errors are adding up. Please provide the errors for all these parameters and estimate the total error when ks is obtained. If this accumulation of errors is large, all the theoretical background introduced in this paper makes little sense.

Author Response

Response to Reviewer 2 Comments

Thank you very much for your extensive work on reviewing our manuscript. We have revised the paper according to your valuable comments. The revised parts of the new manuscript are marked and the problems are addressed below sequentially.

Point 1: What will be the final use of the D-dot probe? Will it be used as an open field sensor? The variation of the D-dot sensor with frequency (see Fig. 1 and Eq. 2) is only valid when the sensor is mounted inside a metallic structure, in respect to which there is a capacitance of the dome. If the sensor is used as an open field sensor this is no more the case.

Response 1: Thank you for pointing out this, this sensor will be finally used for E-field in vicinity of ground plane as a ground-field D-dot sensor. Usually it is used to monitored the applied field in a bounded-wave EMP simulator where an upper plate exists above the ground plane. Therefore, the variation of the D-dot sensor with frequency can be expressed as shown in Figure 2.

Point 2: When the calibration experiments are performed to obtain the calibration factor, there are a large number of parameters that have manufacturing or/and measurement errors that influence the results. For example h in Eq.7, he, C and theta in Eq. 16 are all are known with a certain error. When ks is calculated, these errors are adding up. Please provide the errors for all these parameters and estimate the total error when ks is obtained. If this accumulation of errors is large, all the theoretical background introduced in this paper makes little sense.

Response 2: We fully agree with your comments on the uncertainty of the ks and its influence factors. Therefore, we used an experimental approach to measure the response of the sensor and then obtain the transfer function of it. The OE model can provide a parametric description of the transfer function, it is much clearer than the conventional frequency domain curves. Eq.15~17 here is only used for illustrating the influence factors of ks, the result of the theoretical expression is just used as a reference value rather not a calibrated result here. Relational descriptions have been modified in the new manuscript.

Reviewer 3 Report

Test and calibration are very important for a  physical sensor. A cailbration method based on system identification is proposed in this paper, which has a better performance in sensitivity coefficient calibration as compared to Peak Value Calibration.
I suggest that the draft should be improved in serval sections before being accepted.

1) In section 3, the authors give a calibration experimental system based on TEM cell and corresponding calibration procedure. But in section 4.2, the validaiton experiment is carried on with a GTEM Cell. I think there are some differences, such as the system tranfer function , in those two devices. The authors should explain the equivalence.

2) Figure 10 shows the the waveform of incident electric field and the numerical integration of  D-dot sensor output.  How did the authors get the calibrated result ?  The parameters used in the numberical integration and calibration should be presented. I also doubt the results, as for the response time.  Usually there is an observable time delay between the input and output signal.

3) The authors use a unpublished article in the references ([17]). If the authors have already done some research in frequency domain calibration method for D-dot sensor, they should also compare the proposed calibration method with the frequency domain calibration. At least , they can tell the readers which one is the best in brief.

Author Response

Response to Reviewer 3 Comments

Thank you very much for your extensive work on reviewing our manuscript. We have revised the paper according to your valuable comments. The revised parts of the new manuscript are marked and the problems are addressed below sequentially.

Point 1: In section 3, the authors give a calibration experimental system based on TEM cell and corresponding calibration procedure. But in section 4.2, the validation experiment is carried on with a GTEM Cell. I think there are some differences, such as the system transfer function, in those two devices. The authors should explain the equivalence.

Response 1: The unclear descriptions in original manuscript has been modified. The validation experiment aims to verify the sensitivity coefficient calibrated in Sec. 3. Due to the bandwidth limitation of the TEM cell, the square wave verification experiment carried out on the basis of a GTEM cell. In theory, the sensitivity coefficient should be consistent no matter where it is calibrated, so we used coefficients calibrated in TEM cell to verify the waveforms measured in GTEM cell, which is more able to show the universality of the calibration results.

Point 2: Figure 10 shows the waveform of incident electric field and the numerical integration of D-dot sensor output.  How did the authors get the calibrated result? The parameters used in the numerical integration and calibration should be presented. I also doubt the results, as for the response time. Usually there is an observable time delay between the input and output signal.

Response 2: The setup of the verification experiment based on the calibration platform shown in Figure 3. There are several differences in the experimental setup that needs to be pointed out. First, the SHANGHAI SANKI ENS-24XA high-frequency noise generator was used as an excitation source. Second, due to the bandwidth limitation, the square wave verification experiment carried out on the basis of a GTEM cell. Third, the incident electric field is measured by a high voltage probe, which is placed at the input measurement port of the GTEM cell.

As illustrated in Figure 10, the blue line refers to the incident electric field measured by the high voltage probe, and the red and black line represent to the dE/dt waveform of numerical integration multiplied by the sensitivity coefficient calibrated in Sec 3. Obviously, the coefficient calibrated by system identification method is more accurate. The delay has been removed to facilitate comparison.

Figure 10. Experimental verification of square wave.

Point 3: The authors use an unpublished article in the references ([17]). If the authors have already done some research in frequency domain calibration method for D-dot sensor, they should also compare the proposed calibration method with the frequency domain calibration. At least, they can tell the readers which one is the best in brief.

Response 3: References [17] mainly introduced the frequency domain calibration method based FFT. Due to the oscillation of the transfer function, the FFT method is more difficult to determine the corner frequency and has poorer calibration accuracy. According to your comments and suggestions, we added the comparison of two frequency-domain calibration method (shown in Table 2). It can see that the system identification method is an improvement of the FFT method, which can give a smoother curve of transfer function. In order to avoid confusion, we decided to remove the reference [17].

(a)    FFT method

(b)    System   identification method

Round  2

Reviewer 2 Report

The manuscript is now acceptable for publication.

Reviewer 3 Report

I think main questions have been revised and the authors give relatively clear description about their work, although some improvements are still required in future.